# Exosome-Based Carrier for RNA Delivery: Progress and Challenges

**DOI:** 10.3390/pharmaceutics15020598

**Published:** 2023-02-10

**Authors:** Yicheng Lu, Wei Huang, Meng Li, Aiping Zheng

**Affiliations:** 1Institute of Pharmacology and Toxicology of Academy of Military Medical Sciences, 27 Taiping Road, Haidian District, Beijing 100089, China; 2Zhejiang University-University of Edinburgh Institute (ZJU-UoE Institute), Zhejiang University School of Medicine, 718 Haizhoudong Road, Haining 314400, China

**Keywords:** exosomes, RNA therapy, RNA loading, drug delivery, targeting, progress, challenges

## Abstract

In the last few decades, RNA-based drugs have emerged as a promising candidate to specifically target and modulate disease-relevant genes to cure genetic defects. The key to applying RNA therapy in clinical trials is developing safe and effective delivery systems. Exosomes have been exploited as a promising vehicle for drug delivery due to their nanoscale size, high stability, high biocompatibility, and low immunogenicity. We reviewed and summarized the progress in the strategy and application of exosome-mediated RNA therapy. The challenges of exosomes as a carrier for RNA drug delivery are also elucidated in this article. RNA molecules can be loaded into exosomes and then delivered to targeted cells or tissues via various biochemical or physical approaches. So far, exosome-mediated RNA therapy has shown potential in the treatment of cancer, central nervous system disorders, COVID-19, and other diseases. To further exploit the potential of exosomes for RNA delivery, more efforts should be made to overcome both technological and logistic problems.

## 1. Introduction

The last several decades have witnessed remarkable progress in RNA-based therapeutics for preventing and treating multiple diseases, such as immune diseases, cancers, and infectious diseases [1,2,3]. To date, there is diverse RNA therapeutics potential for disease treatments, including messenger RNA (mRNA), ribosomal RNA (rRNA), microRNA (miRNA), small interfering RNA (siRNA), small non-coding RNA (sncRNA), and long non-coding RNA (lncRNA). These different forms of RNA therapeutics can manipulate genetic or proteomic information that is mediated by distinct mechanisms [4]. However, clinical applications of RNA drugs are primarily tackled by the delivery issue: the instability and hydrophilicity of naked RNA molecules, as well as the lack of efficient carriers for delivering RNA molecules to target cells and tissues [5]. Several studies have shown the unsatisfactory efficacy of delivering naked mRNA [6,7,8], which is caused by the high rates of RNA degradation in circulation, RNA-induced immune response, and poor cellular infiltration [9,10]. Therefore, ideal delivery systems are needed for the safe and efficient delivery of RNA drugs.

Extracellular vesicles (EVs) are lipid bilayer vesicles that are naturally released from different cells [11]. EVs can be divided into three subtypes according to their size: (1) apoptotic bodies (500 nm–2000 nm) are released from the plasma membrane during programmed cell death, (2) microvesicles (100–1000 nm) are also secreted from the plasma membrane, and (3) exosomes (30–150 nm) originate from endosomes [12,13].

Exosomes are small (~30–150 nm) lipid bilayer-coated extracellular vesicles (EVs), which are released into the microenvironment after the fusion of multivesicular bodies (MVBs) with the plasma membranes [14]. Exosomes are actively generated by a variety of cells, including epithelial cells, dendritic cells, mast cells, B cells, T cells, neurons, and tumor cells [15,16,17]. Such vesicles were initially recognized as a minute “cellular dust” extruded on cell storage, or which acted as the mechanism of cellular waste disposal [7]. Moreover, previous research has reported that exosomes are thought to play a crucial role in intercellular communication in various cellular processes, including signal transduction, immunological response, and antigen presentation [10,18,19].

Exosomes have increasingly become an efficient and promising carrier for drug delivery. Compared to lipid nanoparticles (LNP) and other traditional synthesis delivery systems, exosomes have their special advantages. (1) Exosomes have relatively low cytotoxicity and immunogenicity, for they are natural vehicles for cargos like proteins, nucleic acids and lipids, and they have better biocompatibility due to membrane proteins such as tetraspanin and fibronectin [20,21]. (2) Exosomes can circulate in vivo for a longer time than LNPs and display high stability in plasma [22], while LNPs are more likely to be removed by macrophages or reticuloendothelial cells [23]. For example, Kamerkar et al. discovered that, through binding of CD47 on the membranes of exosomes and signal regulatory protein alpha (SIRPα), exosomes can escape clearance by macrophages. They also compared the delivery efficiency of exosomes and LNPs and found out that exosomes showed a longer retention time in circulation [24]. (3) Exosomes can cross the blood-brain barrier (BBB) by a variety of mechanisms [25,26]. In contrast, how to deliver cargos to the central nervous system efficiently with LNPs has always been a tough question. (4) Though, after intravenous injection, unmodified exosomes are likely to be cleared from circulation rapidly and are mainly distributed to the liver, spleen, lungs, and gastrointestinal tract [27,28], by genetic engineering methods such as transfection, homing peptides or ligands on the exosomal membrane can be expressed, which improves targeting and therapeutic efficiency [29,30].

Several studies have reported that exosomes originate from multiple cell lines, and body fluids contain various RNA molecules [31,32], such as miRNA, sncRNA, and mRNA, which can modulate gene expression or be translated to certain proteins in the recipient cells [33,34]. Considering their multi-pronged roles in physical and pathological conditions, interest in exosomes ranges from in vivo features and roles to their potential uses in clinical applications as delivery vehicles, diagnostic particles, and biomarkers [35,36,37]. In recent years, exosomes have been increasingly utilized as a promising targeted delivery system in the growing field of RNA therapeutics for cancers, infectious viruses, and other diseases. Efficient, cost-effective, and streamlined procedures for loading therapeutic molecules into exosomes are prerequisites for translating exosome-based RNA drug delivery into clinical benefits [38].

Although several reviews have summarized the applications of exosomes as a promising carrier for RNA therapeutics [39,40,41], few of them provided a systemic overview of the physiological roles, preparation, and application of exosomes. In this review, we focus on the latest research progress regarding RNA-loading, isolation, and modification of exosomes, as well as their potential to be a functional carrier for delivering RNA molecules in clinical trials. Thoroughly understanding the mechanisms of exosomes as carriers of RNAs may help open up new opportunities for RNA therapeutics.

## 2. Exosomes: Generation, Composition, and Function

### 2.1. Generation of Exosomes

The generation of exosomes starts from the endosomal system, where early endosomes grow into mature endosomes or MVBs, and concomitantly the endosomal membrane invaginates to produce intraluminal vesicles (ILVs) in the lumens of organelles [42].

First, the invagination of the plasma membrane leads to bud formation, during which process fluid, extracellular contents, and cell-surface proteins are incorporated. The budding process gives rise to the de novo formation of early endosomes, or the fusion of the preexisting early endosomes with constituents from the endoplasmic reticulum (ER) and trans-Golgi network (TGN). Early endosomes grow into late endosomes and further form the multivesicular bodies (MVBs) rich with intraluminal vesicles (ILVs) by invagination of the endosomal limiting membrane. In this process, various proteins are sorted into the inward budding endosomal membrane and some of the cytosolic contents are encased within the ILVs. MVBs will then choose between two cell fates. Most of the MVBs are transported to the plasma membrane through the cytoskeletal network, where they release the ILVs (future exosomes) into the extracellular environment upon exocytosis, post fusion with the plasma membrane. Otherwise, MVBs undergo degradation by fusion with lysosomes or autophagosomes [13,14,43,44].

Several mechanisms of the biogenesis have been studied, among which the endosomal sorting complex required for transport (ESCRT)-dependent pathway is most well-known. ESCRT-0 assembles cargos and the ESCRT-I–ESCRT-II complex mediates membrane budding and cargo internalization. Subsequently, ESCRT-III is responsible for the scission of the membrane neck [45].

Once released, exosomes may be taken up by recipient cells in various ways, including endocytosis, receptor-mediated interactions, and membrane fusion [46,47,48]. After entering recipient cells, exosomes are most likely to target lysosomes, where they degrade their cargos [49]. This degradative mechanism could provide metabolites for the recipient cells. Cargos that exosomes deliver can also activate different physiological or pathological reactions in recipient cells [50]. The process of exosome production and cellular uptake is summarized in Figure 1.

### 2.2. Composition of Exosomes

It is the parent cells, environmental stimulation, and physiological conditions that determine the composition and yield of exosomes. Exosomes can contain a variety of proteins, such as membrane proteins, nuclear proteins, and extracellular matrix proteins. Also, cytosolic metabolites and nucleic acids such as mRNA, noncoding RNA, and DNA can be found in exosomes [51,52].

Emerging studies of the exosomal components have displayed the existence of proteins, nucleic acids, lipids, and other components, where lipids are thought to be the main component. There is a glycan canopy attached to the surface proteins and outer leaflet lipids on the outermost surface of exosomes [53]. Beneath this glycan canopy, cholesterol, diglycerides, phospholipids, sphingolipids, glycerophospholipids, and ceramides are enriched in exosomes, relative to the parent cells [21,54]. Also, exosomes contain a wide range of transmembrane proteins, peripherally associated membrane proteins, lipid-anchored membrane proteins, and soluble proteins of the exosome lumen [55]. The most common proteins that can be found in exosomes are Hsp70 and CD63 [56]. It has been proved that certain tetraspanin proteins (CD81, CD82, CD37, CD9, and CD63) are highly concentrated in exosomes [57]. Besides lipids and proteins, functional RNA molecules including mRNAs and other non-coding RNAs are recognized in exosomes [58].

### 2.3. Functions of Exosomes

Traditionally, there are several ways of direct cell-cell communication, such as gap junctions, cell surface protein interaction, or communication with distant cells with the help of secreted factors. Also, chemical and electrical signals participate in intercellular communication [59,60,61]. Emerging studies have elucidated that exosomes carrying function molecules like lipids, proteins, and nucleic acids can act as a mediator of cell-cell communication [62]. Function molecules in exosomes may influence recipient cells through several mechanisms: (1) directly activating target cells via surface ligands; (2) transporting activated receptors to target cells; and (3) reprogramming recipient cells genetically via delivery of lipids, RNAs, and proteins. In this way, parental cells can contact proximal or distal recipient cells [55].

Exosomes play a significant role in immunoregulation, such as immune activation, antigen presentation, immune tolerance, and immune suppression. For example, CD4**^+^**T cells and CD8**^+^**T cells can secrete exosomes that can target dendritic cells (DCs), leading to T cell silence in an antigen-specific way [63]. Exosomes released by regulatory T cells can mediate immune suppression and inhibit Th1 immune response [64]. Regulatory T cells can also secrete CD73-expressing exosomes, which may prevent the activation of CD4**^+^**T cells [65]. Numerous studies have reported that exosomes may have dual functions in regulating tumors, according to their sources and contents. It has been proven that exosomes from cancer cells can stimulate tumor growth as well as tumor immune evasion, as they are enriched with prostaglandin PGE2 and TGF-β [66]. On the other hand, exosomes isolated from dendritic cells can suppress tumor growth and facilitate the antigenic capacity of dendritic cells [67]. Therefore, identifying the most suitable source of exosomes is of great significance in preventing any counterproductive results.

## 3. Preparation of RNA-Loaded Exosomes

### 3.1. Isolation of Exosomes

Almost all normal cells can generate exosomes, such as dendritic cells, human umbilical vein endothelial cells (HUVECs), T cells, B cells, macrophages, mesenchymal stem cells (MSCs), and natural killer (NK) cells [43]. Exosomes can also be found in almost all types of body fluids, such as the blood, serum, synovial fluids, saliva, urine, amniotic fluid, and breast milk [68]. As exosomes are highly heterogeneous in size, density, content, and source, there are still no efficient methodologies for extracting them, and the current exosomal strategies display low purity and high expense. There are several isolation methods widely used in research for different purposes and applications: (1) ultracentrifugation, (2) size-exclusion chromatography, (3) microfluidics, (4) polymer precipitation, and (5) immunoaffinity capture technique (Table 1).

Ultracentrifugation is now considered the gold standard technique for exosome extraction and isolation. Ultracentrifugation mainly isolates the desired components based on the size and density differences of exosomes in solution, so it is generally used for the isolation of large-dose sample components with marked differences in sedimentation coefficient [69]. Though ultracentrifugation is a classical method for exosomal isolation, it is incapable of separating lipoproteins that share similar biophysical properties to target exosomes. Also, ultracentrifugation has problems with time consumption, high expense, structural damage, and lipoprotein co-separation [70,71].

Size-exclusion chromatography is one of the most common techniques for exosomal isolation that isolates components depending on size. Larger molecules cannot pass porous gel filtration polymer, which acts as the stationary phase, while the smaller molecules remain in the gel pores and are eventually enriched in the mobile phase. Size-exclusion chromatography is a rapid, easy, and cost-efficient method, and does not influence the biological properties of exosomes, but may lack specialty and purity [70,72].

Microfluidics-based methods use microfluidic devices to separate exosomes based on physical characteristics such as size and density, or chemical properties such as combining exosome surface antigens. The immune-microfluidic technique, which is similar to the immunoaffinity technique, is most commonly used. Exosomes are isolated by the specific binding to antibodies which are attached to the microfluidic devices as exosome markers. This method makes the separation of exosomes fast and efficient, with high purity. However, the devices needed are complex and expensive [73,74].

In immunoaffinity methods, exosomes are obtained by specifically combining antibodies or magnetic particles. Antibodies are commonly used to recognize specific exosome surface markers, such as the tetraspanins: CD9, CD63, and CD81 [75,76]. Immunoaffinity requires a higher cost to capture exosomes than other isolation techniques, for a large amount of antibody-conjugated beads are needed; immunoaffinity is thus not suitable to produce exosomes with a large amount.

In the polymer precipitation method, polyethylene glycol (PEG) is generally utilized as a medium to gather exosomes by reducing the solubility of exosomes during centrifugation. This method is time-efficient and can deal with large doses of samples. However, this method has the disadvantage of relatively low purity and recovery rate [77,78].

The isolation methods described above are the most commonly used in research, while there are currently more and more novel ways to separate exosomes. In some research, different isolation techniques are combined for a better isolation effect than that of a single technique. For example, combing UC with SEC makes it possible to deal with large amounts of samples as well as increasing the purity of products, which draws on each other’s advantages [79].

Nevertheless, the characterization and quantification of exosomes are still challenges, for exosomes are below the detection range of many current techniques.

To characterize exosomes precisely, size, shape, surface charge, density, and porosity are important properties in the analysis of exosomes. Currently, there are a variety of methods to characterize exosomes, including nanoparticle tracking analysis (NTA) [80], dynamic light scattering (DLS) [81], tunable resistive pulse sensing (TRPS) [82], flow cytometry [83], electron microscopy [84], and atomic force microscopy (AFM) [85]. DLS and NTA can analyze parameters such as size and number. Electron microscopy is commonly used to characterize the structural features of exosomes. Among these techniques, NTA, TRPS, flow cytometry, and electron microscopy techniques have also been commonly used for exosome quantification.

**Table 1 pharmaceutics-15-00598-t001:** Methods of isolating exosomes.

Isolation Methods	Principle	Benefits	Drawbacks	Refs.
Ultracentrifugation	Size, density and sedimentation properties	High purity	Low recovery, large demand of samples, long operation time	[70,71]
Size-exclusion Chromatography	Size	Easy operation, low cost	Low specificity and purity	[70,72]
Microfluidics-based methods	Size and density, or chemical properties	High efficiency, high purity, fast operation	Complexity of device	[73,74]
immunoaffinity methods	Affinity purifications	High purity and specificity	High cost, restriction in the amount of samples	[75,76]
polymer precipitation	Polymer precipitation	Easy and fast operation	Low recovery, low purity	[77,78]

### 3.2. Methods of Loading RNA into Exosomes

It has been proven that exosome RNA can mediate cell-cell communication. Initial research showed the presence of approximately 121 miRNAs that were found in exosomes, some of which were expressed at a higher level than that in donor cells [34]. Therefore, exosomes may be ideal natural delivery vehicles for RNA drugs in gene therapy. Despite various advantages, how to produce adequate RNA-loaded exosomes can be a limiting factor for their in vivo application. So far, several methods have been proposed to incorporate RNAs into exosomes.

The most straightforward way to load drugs into exosomes is incubation. In this way, drugs are incubated with exosomes or the donor cells at a specific temperature and then diffuse into exosomes by a concentration gradient. Incubation has been widely used to load hydrophobic cargoes like curcumin and paclitaxel, since the plasma membrane of exosomes are hydrophobic and lipid-enriched [86,87]. However, the hydrophilicity of nucleic acids may lead to unsatisfying loading efficiency. Bryniarski et al. packaged miR-150 mimics into T-cell-derived exosomes by direct incubating for 1 h [88]. To enhance loading efficiency, Gong et al. modified miR-159 with cholesterol and then incubated miR-159 with macrophage-derived exosomes to acquire miRNA-loaded exosomes [89].

Transfection is the most common method for loading RNA into exosomes. To generate RNA-loaded exosomes, specifically encoded plasmids are transduced into the exosome-generating cells. In this way, the desired RNA can be sorted into exosomes in donor cells through natural mechanisms [90,91]. Although several natural mechanisms have been demonstrated regarding the sorting of exosomal RNAs, the exact mechanisms of the process still remain quite elusive. Some studies have reported that short nucleotide sequences of RNAs may guide their transport to different locations, including exosomes [92,93]. Besides specific recognition motifs, the secondary/tertiary structures of RNAs [94] and lipids involved in the process of exosome generation may play an important role as well in the sorting process. It has also been discovered that RNAs transfer from the nucleus to specific locations with the help of RNA-binding proteins (RBPs), such as AGO2 [95], ALIX70 [96], annexin A2 [97], major vault protein (MVP) [98], HuR75 [99], etc. siRNA and miRNA can directly transfect donor cells and then be packaged into exosomes naturally. In addition, commercialized reagents can be used in transfection. For example, after mixing siRNA AF488 with lipofectamine and exosomes isolated from HeLa cells, exogenous siRNA can be loaded into exosomes, which can further transfect recipient cells via incubation [100]. Though convenient, this method exhibited the disadvantages of cytotoxicity, inefficient packaging, and low specificity [101]. In addition to transferring specific molecules into exosome-secreting cells and exosomes, another strategy is to express RNA-binding proteins on the surface of exosomes through transfection. The efficiency of loading desired RNA into exosomes can be improved by RNA-binding domains binding to specific motifs of RNA [102]. Further studies are needed to verify whether the binding of RNAs to their RNA-binding domains influences their biological functions.

Physical methods such as electroporation [103,104], sonication [105], and dialysis [106], can be utilized to load RNAs into exosomes by creating micropores on the membranes of exosomes, or undermining the integrity of the membranes [107]. These methods are commonly used to package short RNAs like siRNA and miRNA. Though using physical methods to load RNA is highly efficient, it may lead to damage to the integrity of the lipid bilayer, the aggregation of cargos and exosomes, and the inactivation of RNAs [108]. Therefore, the experimental parameters of these physical methods need to be accurately controlled to reduce the aforementioned risks.

In addition to the classical methods mentioned above, there are several other potential ways to load RNAs into exosomes. For example, the Exo-Fect™ Exosome Transfection Kit that was recently developed by System Biosciences Co. for exosome delivery enables the insertion of RNAs, DNAs (including plasmids), and small molecules into isolated exosomes [109]. Also, cellular nanoporation methods have been used to induce the generation of RNA-loaded exosomes by activating cells with electrical motivations [110]. So far, it is somewhat feasible to encapsulate small RNAs like siRNAs and miRNAs into exosomes. However, though various methods have been developed, improving the efficiency of mRNA loading is still a challenge.

The aforementioned techniques for RNA loading, along with their principles, benefits, and drawbacks are summarized in Table 2.

## 4. Targeted Delivery of Exosome Surface Modification

Most studies regarding exosome biodistribution indicated that natural exosomes spread in the extracellular fluid through free diffusion and are taken up by recipient cells through different mechanisms. To deliver specific cargos to target cells or tissues, multiple exosomal engineering techniques and corresponding delivery strategies have been developed (Figure 2), which are indispensable for practical applications since they confer exosomes with extra functionality. Here we mainly described strategies that are commonly used for the targeted delivery of exosomes.

### 4.1. Receptor Targeting

Exosomes are generated by multiple types of cells and carry donor-cell-specific ligands, lipids, and adhesion molecules. Vectors that can trigger the production of ligands on exosomal surfaces or fusion proteins associated with ligands have been developed. To date, the ligand-receptor-binding-based targeting strategy has been the most commonly used, and is a highly specific method for delivering exosomes that specifically bind to receptors on the cells of interest. In order to introduce the desired ligands on the exosomal surface, there are currently two common approaches: (i) ligand modification through transfection and (ii) direct chemical assembling of ligands.

#### 4.1.1. Transfection-Based Ligand Modification

Vectors that produce specific ligands on the exosomal surface or fusion proteins associated with ligands have been developed [111,112]. The capacity for targeted delivery of exosomes was first evaluated on cancer cells. Epidermal growth factor receptor (EGFR), one of the growth factor-binding receptors, has been demonstrated to be overexpressed in multiple epithelial human cancer cells that served as a bull’s-eye for cancer drug delivery [113,114,115]. In 2013, Ohno et al. engineered human embryonic kidney cell line 293 (HEK293) to present the transmembrane domain of platelet-derived growth factor receptor (PDGFR) that fused to the GE11 peptide (amino-acid sequence YHWYGYTPQNVI) [112]. The PDGFR transmembrane domain facilitates GE11 peptide expression, which specifically recognizes and binds to EFFRs on the surfaces of exosomes. Intravenous injection of the exosomes on recombination-activating gene 2 (RAG2) knockout mice revealed the targeted delivery of let-7a miRNA to EGFR-expressing xenograft breast cancer tissues. The other receptor whose expression is increased in cancers, including gastric and breast cancers, is the human epidermal growth factor (HER2) [116,117]. In the study conducted by Limoni et al., HEK239T cells were transduced by a lentiviral vector that carried the pLEX-LAMP2b-DARP, then the secreted exosomes were isolated, characterized, and incubated with siRNA for subsequent delivery [118]. The results indicated that the exosomes with DARPin G3 presented on the surface exhibited elevated affinity to HER2/Neu-positive breast cancer cells.

In addition to cancer therapy, the demand for targeted drug delivery to normal cells is increasing due to the increase in nervous system disorders and other diseases, particularly those protected by physiological barriers such as the blood-brain barrier (BBB). Alvarez-Erviti et al. engineered dendritic cells to express Lamp2b that fused to the neuron-specific rabies viral glycoprotein (RVG) peptide through transfection [104]. Exosomes from these cells were isolated, purified, and loaded with siRNA, which can specifically bind to acetylcholine receptors in the central nervous system (CNS) and deliver cargos.

#### 4.1.2. Direct Chemical Installation of Ligands

Although transfection is effective for presenting genetically engineered proteins on the exosomal surface, it also exhibits several limitations, such as being expensive and complicated, as well as the inability to carry molecules other than genetically encodable peptides and proteins. The other method for adding desired ligands to the exosomal surface is direct chemical assembling, which allows for applying different types of ligands on the exosomal membrane through hydrophobic insertion or conjugation [119,120,121]. For delivering chemotherapeutic drugs in cancer treatment, Wang et al. chemically labeled human umbilical endothelial cells with dual ligands, including biotin and avidin, in the phospholipid membrane, which can produce dualligand-conjugated exosomes with strong abilities to target tumor cells [122]. The exosomes then deliver drugs into cancer cells that can induce apoptosis. Studies have also evaluated potential applications in the treatment of non-cancerous diseases. It has been reported that the natural regenerative exosomes, such as exosomes derived from cardiac stem cells or bone marrow mesenchymal stem cells (BM-MSCs), can be specifically delivered to femur fracture regions or myocardial infarction sites through chemically conjugating with stromal cell-specific aptamer or cardiac homing-peptides [123,124].

### 4.2. pH Gradient Guiding

The specific chemical features of different cells or tissues are considered a focus for developing novel exosomal targeted delivery. Compared with normal cells, cancer cells display a higher rate of glycolysis and thereby elevated lactate production, contributing to an acidic microenvironment [125,126]. This property can be used for building a pH-responsive delivery system to specifically target cancer cells and reduce therapeutic side effects. Lee et al. reported engineered exosomes carrying the hyaluronic acid grafted with 3-(diethylamino) propylamine (HDEA) that can efficiently respond to the pH of the cancer microenvironment and specifically bind to CD44 receptors, thus alleviating tumor growth in vitro [127].

### 4.3. Magnetic Force Directing

In addition to the targeted delivery approaches that use either biological or physicochemical properties of certain cells or tissues, the delivery can also be achieved by utilizing external attractive forces like magnetism [128,129,130]. In this methodology, exosomes are installed with magnetic compounds, of which the isolation and movement are guided by the force of an external magnetic field. In the study conducted by Qiu et al. [131], a dual-functional exosome-based superparamagnetic nanoparticle cluster was designed as a therapeutic carrier for cancer therapy. The exosomes displayed a strong response to the magnetic field at room temperature, thus allowing the exosomes to be isolated from blood and delivered to cancer cells [131]. Moreover, Qiu et al. reported that these exosomes can deliver doxorubicin to cancer cells and suppress cancer growth.

Taken together, various methodologies have been developed for exosome engineering to efficiently deliver cargos to the target. Moreover, a combination of exosomal modification techniques can be used to achieve highly specific and more efficient drug delivery.

## 5. The Application Potential of RNA-Loaded Exosomes

With the advantages of safety, high biocompatibility, high bioavailability, and the ability to penetrate the BBB, exosomes have become increasingly promising and potential carriers for RNA drug delivery.

Exosomes can be found in almost all kinds of body fluids, such as the blood, serum, synovial fluids, saliva, urine, amniotic fluid, and breast milk [68]. They can be transported passively throughout the human body, while their target delivery is mainly associated with their surface target molecules from parent cells [104]. There are now various novel exosome bioengineering approaches that aim to improve exosome targeting delivery for both research and treatment purposes. So far, exosome-mediated gene therapy is currently mainly used for the treatment of cancer and brain diseases. Furthermore, exosome-mediated RNA therapies for other diseases such as infectious, ocular, liver, and kidney diseases have also been developed.

### 5.1. Oncology Therapy

To improve the effectiveness of cancer therapy, it is urgent to deliver drugs to tumor cells specifically and accurately. Several RNA cancer therapies are in various stages of development, including mRNA, tRNA, siRNA, and RNA interference (RNAi) therapies, which are mainly concentrated in the early preclinical and clinical stages. The key to promoting RNA cancer treatment is to establish an effective delivery system that can solve the problems of early degradation, toxicity, and non-target-specificity of RNA molecules. Compared with other nanoparticle drug-delivery systems, exosomes have higher bioavailability for tumor tissues, as well as lower cytotoxicity and immunogenicity for normal tissues [132]. Also, the transmembrane- and membrane-targeting proteins of exosomes may help to promote the uptake of their loads [95,96,97,98,99]. For example, Kim et al. discovered that, in comparison with paclitaxel-loaded liposomes, paclitaxel-loaded exosomes achieve higher cell uptake and better therapeutic qualities in mouse lung cancer models [133]. After packaging RNA drugs into exosomes, they can directly target cancer or cancer stem cells’ (CSC) specific signaling pathways to prevent the growth of tumors. In addition, ligands of exosomes that react with receptors of cancer cells can be modified to promote the cellular uptake of vaccines by cancer cells. So far, researchers have used exosomes as nanocarriers to load mRNA, miRNA, siRNA, and circular RNA (circRNA) for cancer gene therapy.

mRNA cancer drugs can treat cancers by encoding and expressing tumor-specific antigen (TSA), a tumor-associated antigen (TAA), and other cytokines. It has been proven that exosomes may be a potential delivery carrier to increase the stability and transfection efficiency of mRNA as well [134]. Wang et al. utilized HEK-293 and dendritic cells from mice to produce exosomes that loaded HChrR6-encoding mRNA (prepared by transfecting cells with encoding plasmid) and then delivered it to HER2+ human breast cancer cells. As a bacterial enzyme, HChrR6 can transform the predrug 6-chloro-9-nitro-5-oxo-5H-benzo-(a)-phenoxazine (CNOB) into the drug 9-p amino-6-chloro-5H-benzo[a]phenoxazine-5-one (MCHB) in tumors. The use of exosomes loaded with HChrR6 mRNA can target HER2 receptors (EXO-DEPTs) with CNOB. The results showed that HER2+ cells could be specifically killed and the development of HER2+ human breast cancer in athymic mice models could be nearly stopped [135]. Forterre et al. subsequently proposed a similar way, with better safety, for breast cancer therapy. In the research, in-vitro-transcribed (IVT) HChrR6 mRNA was packaged into exosomes to eliminate the potential risk of harmful plasmid transfection of the exosome donor cells used in the previous study. Additionally, CB1954 was utilized as a prodrug that HChrR6 could activate, as its safe human dose was already clear [136]. Besides breast cancer, exosome-based mRNA delivery has also shown a therapeutic benefit in leukemia. Usman et al. separated exosomes from human red blood cells (RBCs) to treat acute myeloid leukemia (ALL) cells. Cas9 mRNA and guide RNA were loaded into exosomes and then targeted the human miR-125b-2 locus. The results demonstrated that RBC extracellular vesicles can deliver the CRISPR-Cas9 system to leukemia cells and decrease the expression of miR-125a [137]. Yang et al. modified exosomal CD47 with glioma-targeting peptides to improve the cellular uptake of exosomes in glioma cells. This research indicated that mRNA-containing exosomes can transfer tumor suppressor genes into glioma cells and restrain the growth of tumors [110]. In another study regarding Schwannoma treatment, exosomes containing CD-UPRT mRNA were released after donor cells were transfected with a plasmid encoding a cytosine deaminase (CD) fused with uracil phosphoribosyltransferase (UPRT). The results showed that direct intratumoral administration promoted the conversion of the prodrug 5-fluorocytosine (5-FC) to 5-fluorouracil (5-FU) within tumor cells and significantly regressed the growth of tumors [138].

miRNAs are a type of ncRNA commonly found in the non-coding region, which do not encode proteins. miRNAs play a significant role in regulating gene expression, since they can target mRNAs and promote their degradation or facilitate translation inhibition. The level of different miRNAs can thus efficiently affect the occurrence and growth of tumors. For instance, miR-708 can restrain the proliferation, survival, and migration of lung cancer cells, which is partly attributed to the inhibition of PGE2 signaling. In contrast, overexpression of miR-411-5p/3p may accelerate the proliferation and migration of non-small-cell carcinoma cells and inhibit cell apoptosis [139,140]. There are mainly two strategies of miRNA-based therapy: miRNA promotion and inhibition. The former utilizes exogenous miRNA mimics to increase the level of miRNAs that can prevent the development of tumors. For example, it has been proven that miR-375 is inversely related to the process of epithelial-mesenchymal transition (EMT) in cancer patients and can prevent tumor cells from invasion and migration. Rezaei et al. separated exosomes from HT-29 and SW480 and then loaded them with miR-375 mimic. The results implied that miR-375-loaded tumor exosomes could effectively target metastatic colorectal cancer tissues and control the development of tumors [141]. The latter strategy delivers specific miRNA inhibitors or AMOs to prevent miRNA from promoting the growth of tumors. Naseri et al. derived exosomes from bone-marrow-derived mesenchymal stem cells (MSCs-Exo) and LNA (locked nucleic acid)-modified anti-miR-142-3p oligonucleotides were loaded to downregulate the level of miR-142-3p and miR-150 in 4T1 and TUBO breast cancer cell lines. The results implied that MSCs-Exo can penetrate tumor tissues efficiently and regulate the expression levels of miRNAs of cancer cells [142]. In order to improve the efficiency of miRNA therapy against cancers, some researchers combine gene therapy with chemotherapy. It has been proven that miRNAs and chemotherapeutic drugs can be co-loaded into engineered exosomes to enhance therapeutic qualities. Liang et al. utilized engineered exosomes to encapsulate 5-FU and miR-21 inhibitor oligonucleotide (miR-21i) together to target 5-FU-resistant colorectal cancer cell line HCT-1165FR. Consequently, the co-delivery system significantly improved the efficiency of 5-FU in drug-resistant colon cancer cells and downregulated miR-21 expression simultaneously [143].

siRNAs are another kind of RNA with double strands. siRNAs are mainly involved in RNAi reactions and regulate gene expression by inducing gene silencing. It is known that the development of cancer is closely associated with the overexpression of anti-apoptotic proteins such as KRAS, BCL-2, PLK1, survivin protein, and cell growth factors [144]. Therefore, siRNA cancer therapy currently focuses on reducing the expression level of these oncogenes. The use of siRNAs to modulate gene expression provides new ideas for cancer therapy, but safe and efficient drug delivery systems are still needed to fully realize their clinical potential. As a natural transporter, exosomes are ideal carriers for siRNAs with less toxicity and immunogenicity. To treat pancreatic cancer, Sushrut Kamerkar et al. isolated exosomes from normal fibroblast-like mesenchymal cells to load siRNA to specifically target oncogenic KRASG12D (iExosomes). The exosomes were modulated with CD47 to prolong their retention time in circulation. The results indicated that iExosomes can efficiently target tumor cells and suppress pancreatic cancer in mouse models [24]. Greco et al. used separated exosomes from HEK293 cells and MSC to deliver PLK-1 siRNA targeting bladder cancer tissues, resulting in cell apoptosis by downregulating the expression of PLK-1 [145]. Similar to miRNAs, siRNAs can be used with chemotherapeutics to overcome drug resistance. siRNA increases the cytotoxicity of drugs to cancer cells by silencing genes promoting drug resistance. For example, Li et al. utilized exosomes to deliver siRNA specific to Grp78 together with Sorafenib, resulting in the reduction of drug resistance in sorafenib-resistant cancer cells [146]. Similarly, Lin et al. used iRGD-modified exosomes to load siCPT1A, which can silence the gene expression of fatty acid oxidation (FAO) and promote drug resistance [147]. In addition, siRNA can be used for cancer immunotherapy which utilizes the patient’s immune system to kill cancer cells. Some researchers have designed siRNA-loaded exosomes to target the tumor microenvironment (TME) or immune checkpoints. Lin et al. modified exosomes with PD-1 antibodies and loaded siRNA into them to target A549 and H460 cells, leading to cell apoptosis of cancer cells [148]. Zhou et al. loaded oxaliplatin (OXA) combined with gal-9 siRNA into exosomes isolated from BM-MSCs. The co-delivery of these two drugs can induce tumor-suppressive immune reactions by polarizing anti-tumor macrophages, recruiting cytotoxic T lymphocytes, and downregulating Tregs [149]. In general, these methods are promising for developing effective cancer therapy based on siRNA-loaded exosomes.

Circular RNAs (circRNA) are an important class of noncoding RNAs without a 5′ cap or 3′ poly (A) tail. Through back-splicing by RNA polymerase II, circRNAs are covalently closed RNA circles. circRNA has been proven to participate in various biological reactions, such as cell proliferation, apoptosis, and survival. Also, some disease-related circRNAs can bind specific proteins, regulate RNA editing and transcription, serve as specific microRNA sponges, or produce peptides through translation [150,151]. Accumulating evidence indicates that circRNAs play a role in various biological responses in the TME, involving immune escape, drug resistance, tumor metabolism, the EMT, and angiogenesis. Despite the uncertainty of the function and mechanism of circRNAs in cancer cells, an increasing number of scientists have paid attention to circRNAs in the research fields of lung, breast, gastric, and colorectal cancer [152]. Although most of the research regarding circRNAs focuses on targeting natural exosomal circRNAs, some cases investigate exosome-based delivery of circRNAs. For example, Guo et al. identified a novel circRNA (circDIDO1) that significantly repressed gastric cancer progression. They designed circDIDO1-loaded RGD-modified exosomes (RGD-Exo-circDIDO1) to modulate the miR-1307-3p/SOSC2 axis, leading to the suppression of gastric cancer growth [153]. In general, various natural exosomal circRNAs participate in the progression of cancers and provide clues for targeted cancer therapeutics.

In addition to RNAs mentioned before, lncRNA, short hairpin RNA (shRNA), aptamer, and other types of RNA have gradually been introduced and explored to treat cancer. Exosomes can also act as suitable and efficient carriers for these RNAs. For example, Xiong et al. discovered a cancer-specific aberrant transcription-induced chimeric RNA encoding A-Pas, a cancer-specific chimeric protein that can modulate the occurrence and EMT of tumors. They used exosomes isolated from chimeric RNA-transfected dendritic cells to construct antitumor vaccines to treat esophagus cancer, resulting in the suppression of cancer and prolonged survival time of mouse models with tumors [154]. Huang et al. prepared the c(RGDyK)-modified engineered exosomes and packaged lncRNA MEG3 into exosomes. The lncRNA-loaded exosomes can be efficiently delivered to osteosarcoma cells and facilitated therapeutic effects [155].

The previously described applications of RNA-loaded exosomes in cancer therapy are listed in Table 3. Taken together, exosomes show great advantages as drug delivery carriers for RNA cancer therapy, for they can prolong the retention time of nucleic acids and enhance the target tumor cells efficiently due to their low immunogenicity and low toxicity. More effort needs to be made to fully realize the potential of RNA-loaded exosomes to promote their clinical application in the field of cancer treatment.

### 5.2. Infectious Disease Vaccines

Vaccines for infectious diseases are currently one of the most developed fields of gene therapy. To date, most studies use mRNA as therapeutic molecules to construct RNA vaccines. mRNA vaccines are taken up by target cells and then translated into antigens to induce immune responses. The key to delivering mRNA is to construct an efficient and stable delivery system. So far, most mRNA vaccines for research, or clinical use, use lipid nanoparticles (LNP) and virus-like particles (VLP) as delivery vehicles [156,157]. However, these vectors have problems with cytotoxicity and immunogenicity [158]. Exosomes are thought to have the ability to overcome these limitations.

Accumulating evidence has indicated that exosomes play an important role in cell-cell communication and regulation. Exosomes are involved in the pathogenesis of many infectious diseases. Pathogens can be transmitted through incorporation into exosomes produced by host cells, or by viruses, parasites, and bacteria [159]. During the process of infection, exosomes can convey pathogen molecules such as nucleic acids, proteins, carbohydrates, and lipids that induce host defense and immunity, or mediate immune evasion. Pathogen-loaded exosomes can act as antigen vehicles which may induce acquired immune responses. They can mediate T cell activation directly by carrying processed antigens, major histocompatibility complex (MHC) molecules, and costimulatory molecules [160]. These exosomes with pathogens can also suppress immune responses through several mechanisms. For instance, exosome loading microorganisms such as Human Immunodeficiency Virus (HIV) Nef or Leishmania gp63 can inhibit T cell activation and lead to apoptosis of immune effector cells [161,162]. In addition, exosomes from infected cells may enhance the infection of noninfected cells [163]. The important functions of exosomes in the course of infection offer new ideas for the treatment of infectious diseases. Compared with other nanocarriers, the use of exosomes for infectious diseases has a variety of advantages, including (1) optimized biodistribution due to their ability to circulate in the body; (2) stable environment for therapeutic molecules like RNA; (3) efficient cellular uptake by APCs; (4) the natural role of transport antigens between cells.

The outbreak of COVID-19 catalyzed the rapid development of mRNA vaccines. With the approval for use in humans of the Pfizer–BioNTech vaccine BNT162b2 and the Moderna vaccine mRNA-1273, an increasing number of researchers have realized the potential of mRNA vaccines and devoted themselves to the development of new mRNA vaccines [164]. However, unlike other nanocarriers, currently only a very small portion of researchers have paid attention to exosomes as RNA carriers for infectious diseases, as the mechanism and potential of exosomes are still unclear. Several studies have utilized exosomes to construct COVID-19 mRNA vaccines. For example, Tsai et al. derived exosomes from 293F cells and loaded mRNAs encoding SARS-CoV-2 spike and nucleocapsid proteins into exosomes. The results indicated that the mRNA-loaded exosomes can both induce humoral and cellular immune responses. Meanwhile, they compared the efficiency of delivery between exosomes and LNPs by loading Antares2 mRNA into the two carriers, respectively. After co-incubation with human cells, the Antares2 luciferase activity of each group was examined. The results implied that mRNA-loaded exosomes promoted a higher level of expression than mRNA-loaded LNPs [165].

Although there are various examples of using exosomes to deliver pathogens such as proteins in the treatment of infectious diseases, RNA-loaded exosomes have not been widely used. However, many studies have shown that the use of exosome-mediated RNA therapy for infectious diseases is feasible and promising, and further studies are needed to expand knowledge in this field.

### 5.3. CNS Disease Therapy

In addition to playing an important role in intercellular communication, exosomes can also maintain myelin sheath, mediate neuronal protection, eliminate waste, and promote nerve and tissue regeneration in the central nervous system [166]. Delivering drugs to the CNS is notoriously difficult, as the BBB is impenetrable to most therapeutic agents. It has been proven that cell-derived exosomes or bioengineered drug-loaded exosomes can penetrate the BBB and can be stable in peripheral circulation [167]. The special characteristics of exosomes to deliver drugs provide novel therapeutic strategies to treat central nervous system diseases. Exosomes have been used to treat stroke, dementia, Alzheimer’s disease (AD), Parkinson’s disease (PD), Huntington’s disease (HD), traumatic encephalopathy (CTE), and amyotrophic lateral sclerosis (ALS).

Parkinson’s disease (PD) is a common neurodegenerative disease resulting from an imbalance between excitatory and inhibitory neurotransmitters in the substantia nigra of patients [168]. It is known that catalase can alleviate neuroinflammation caused by reactive oxygen species generated by neurotoxic reagents such as 6-OHDA [169]. Kojima et al. designed a set of devices called EXOsomal transfer into cells (EXOtic) for HEK-293 cells to produce exosomes containing catalase mRNA. The results indicated that catalase mRNA-loaded exosomes could efficiently attenuate neuroinflammation induced by 6-OHDA in mouse models [170]. Esteve et al. revealed the potential of exosomes isolated from cord blood mononuclear cells as a delivery vehicle of miR-124-3p to treat PD. miR-124-3p has been recognized as a promising therapeutic molecule for PD on account of its neurogenic and neuroprotective effects. The results showed that miR-124-3p-loaded exosomes promoted neuronal differentiation of neural stem cells and the protection of N27 dopaminergic cells against 6-OHDA-induced cytotoxicity [171].

Alzheimer’s disease (AD) is a prevalent neurodegenerative disease among the elderly. Currently, the cause of AD remains unclear and it is commonly assumed that amyloid-beta (Aβ) plaques and neurofibrillary tangles (NFTs) are related to the pathophysiology of AD [172]. Due to the multiple valuable functions of exosomes, they have been applied to treat AD as drug delivery carriers. In 2011, Alvarez-Erviti et al. first used RNA-loaded exosomes to treat AD. They derived exosomes from dendritic cells and bioengineered exosomes to express Lamp2b to target neuron-specific RVG peptides. GAPDH siRNA was packaged into RVG-targeted exosomes and then delivered to the brain, leading to a powerful mRNA and protein suppression of BACE1, a therapeutic target in AD [104]. In another work, miR-124a was loaded into exosomes and delivered to astrocytes, resulting in the enhancement of the expression of excitatory amino acid transporter 2 (GLT1) and glutamate uptake of them. In this way, neuronal apoptosis in AD is expected to be suppressed [173].

Huntington’s disease (HD) is a mostly inherited neurodegenerative disease. A mutation results in the expression of an abnormal mutant protein (mHtt), which may be chronically harmful to brain cells via various possible mechanisms [174]. According to previous findings that miR-124 is one of the most crucial miRNAs in the pathology of HD, Lee et al. generated miR-124-loaded exosomes (Exo-124) from transfected HEK 293 cells. After the injection of Exo-124 into the striatum of HD mice, the expression of REST protein (the target protein of miR-124) was lower than that of the control group. Though the behavior of the HD mouse did not remarkably improve, the study provided proof for exosome-mediated delivery of miRNAs to the CNS [175].

A common feature of these neurodegenerative diseases is the deposition of insoluble proteins. After the RNA-loaded exosomes are delivered to the CNS, they can regulate the expression of these key proteins through transcription, gene silencing, and gene knockout to achieve therapeutic effects. Besides neurodegenerative diseases, RNA-loaded exosomes can be applied to treat stroke. After a stroke, the BBB is disrupted. It has been proven that exosomes can contribute to reconstruction in the long term by regulating peripheral immune response, promoting angiogenesis, and enhancing axonal dendritic remodeling [176]. As the miR-17-92 cluster has been identified to promote oligodendrogenesis, neurogenesis, and axonal outgrowth, Xin et al. derived miR-17-92 loaded exosomes (Exo-miR-17-92+) from transfected MSC. After the intravenous administration of Exo-miR-17-92+ in stroke mouse models, there were obvious effects on the enhancement of neurological function and neurogenesis. In addition, the treatment significantly repressed the expression of PTEN, a miR-17-92 target gene, resulting in the promotion of the phosphorylation of PTEN downstream proteins, such as Akt, mTOR, and GSK-3β [177]. In another similar work, miR-133b loaded-exosomes were generated from MSCs to treat stroke. The results showed that the neurite outgrowth poststroke was promoted by stimulating the secondary release of astrocytic exosomes [178].

Exosomes can be used both as diagnostic indicators and as a means of treatment for central nervous system diseases. Exosomes are currently considered promising vehicles for transporting therapeutic molecules like RNAs to previously inaccessible brain areas due to their ability to penetrate BBB.

### 5.4. Other Potential Applications

Exosome-mediated RNA therapy has shown potential in the treatment of liver diseases. For example, Herrera et al. discovered that exosomes with cellular RNA separated from human liver stem cells may promote the proliferation of liver cells [179]. Zhao et al. found that hepatitis B virus (HBV) miR-3-loaded exosomes separated from HBV-infected hepatocytes could induce an innate immune response to inhibit HBV replication by multiple mechanisms [180]. Accumulating evidence has indicated that exosomes can serve as an efficient carrier for therapeutic RNA to treat liver diseases.

Exosomes can also serve as a promising carrier in the treatment of kidney diseases. It has been proven that miRNA can suppress the process of kidney fibrosis by regulating the level of collagen and extracellular matrix [181]. Thus, the delivery of RNA mediated by exosomes may provide new ideas to treat kidney fibrosis. For instance, Wang et al. constructed bioengineered MSCs that overexpressed miR-let7 to target kidney tissues. These MSCs could secrete active miR-let7c-loaded exosomes to alleviate fibrosis by inhibiting the expression of TGF-βR1 protein [182].

In the field of the treatment of lung diseases, exosomes can be a potential carrier for RNA therapy. Exosomes have the potential to overcome difficulties in delivering therapeutic molecules to lung tissue. Zhang et al. derived exosomes from serum and loaded small RNA into them. The results showed that the delivery system could efficiently target lung macrophages, suggesting that therapeutic RNA-loaded exosomes could be an effective and promising tool to treat inflammatory lung responses [183].

## 6. Challenges and Perspectives

Exosomes have been demonstrated to play a crucial regulatory role in cell-cell communication due to their capacity to deliver functional cargo molecules to recipient cells through circulation, and reprogram physiological processes in these cells. Exosomes are therefore promising therapeutic carriers in cell-free strategies for a variety of diseases due to their immunomodulatory properties. There are several advantages of utilizing exosomes as cargo delivery vehicles compared with liposomes and nanoparticles, particularly their stability in circulation. Covered with a biofilm, exosomes are capable of avoiding degradation by macrophages, which provides prolonged circulation times [184,185]. However, the application of exosomes in clinics is still in its infancy. There are a number of challenges that hinder the therapeutic application of exosomes.

The first and paramount challenge is the large-scale production of exosomes with high quality and well characterization [186,187]. Exosomes isolated from either biological fluids or cell culture supernatants are low in yield [184,186,188]. Progress in exosome production has been made with the adoption of various techniques, including 3D scaffolds, bioreactors, and microfluidics-based technologies. The exosome yield can be increased up to 10-fold in a bioreactor, compared with conventional flasks [189]. To achieve clinical application, current methods for exosome isolation and purification still need to be optimized and streamlined for efficient, cost-effective, and reproducible production [190,191]. Furthermore, it is expected to be able to obtain exosomes derived from different types of biological sources by a single exosome isolation technique based on the hyphenation of current strategies. Current techniques used for exosome characterization and validation, including transmission electron microscopy (TEM) and fluorescence-activated cell sorting (FACS), cannot independently analyze the biochemical and biophysical features of exosomes [184,187,192]. More sophisticated and definite techniques are needed to characterize exosomes in clinics.

Techniques are needed to artificially increase the contents of specific RNAs [106]. Nevertheless, exogenous RNA insertion in cell-derived exosomes results in low yield, particularly with large mRNAs [110]. The second challenge is to enhance the efficiency of loading RNA into exosomes and facilitate their targeting efficiency. Current methodologies for exosome-nucleic acid loading including transfection, incubation, and physical treatments exhibit low efficiency [106,193,194]. In particular, electroporation has been reported as the best approach for exosome loading [107,184,195,196]. However, this approach can lead to adverse alterations of exosomal cargos, such as the aggregation and degradation of nucleic acids [197,198]. To increase therapeutic RNA levels in exosomes, efficient new loading strategies should be developed. Recently, Yang et al. developed a cellular-nanoporation (CNP) approach for loading therapeutic mRNAs and peptides into exosomes [110]. Cultured on a specially designed CNP biochip, cells were transfected with plasmid DNAs and triggered to secrete exosomes carrying transcribed mRNAs by a transient and focal electrical stimulus. Compared with bulk electroporation, this method yields up to 50-fold more exosomes, which contain 103-fold higher levels of mRNA transcripts. These approaches are promising to enhance the efficient loading of RNAs and facilitate the therapeutic use of exosomes.

The third challenge is to evaluate the safety, immunogenicity, and biodistribution of therapeutic exosomes in clinical trials. Since exosomes carry a group of signaling molecules, the administration of exosomes may elicit immune responses in the host, contributing to the toxicity and rapid clearance of exosomes [193,199]. The source of the donor cell and the composition of exosomes can both determine its toxicity and immunogenicity. Different types of immortalized cell lines, such as HEK293T cells, have been widely used for the production of exosomes due to the infinite production of exosomes and the ease of genetic modification [200,201,202]. Concerns have been raised that exosomes derived from immortalized cells may carry toxic or even carcinogenic components, and thus should not be used for producing therapeutic exosomes [203,204]. Therefore, a series of preclinical studies such as pharmacodynamics and toxicity examination need to be performed to prevent potential safety concerns and minimize side effects. Utilizing autologous exosome for therapeutic drug delivery has the potential to prevent unpredictable immunogenic concerns in clinics.

The clinical translation of exosomes is an exciting but complicated process filled with numerous challenges. For the successful commercialization of exosome-based therapeutics, not only scientists but commercial manufacturers should put effort in relevant investigations to resolve both technological and logistic issues.

## 7. Conclusions

Taken together, accumulating evidence has demonstrated the potential of exosomes in delivering RNA. Exosomes have low immunogenicity, high biocompatibility, and high delivery efficiency, providing new ideas for the delivery of therapeutic RNA molecules. There are many methods for loading RNA into exosomes, such as incubation, transfection, and physical methods. However, further studies are still needed to achieve satisfactory loading efficiency. To improve the delivery efficiency of exosomes, various exosomal surface engineering techniques have been developed. Though exosome-mediated RNA therapy is considered a promising approach for the treatment of many diseases, many challenges remain to be overcome, such as the large-scale production of exosomes, the insufficient efficiency of RNA loading, and the evaluation of therapeutic effects. Further studies are needed to solve these problems and realize the full potential of exosome-based RNA therapy.

## Figures and Tables

**Figure 1 pharmaceutics-15-00598-f001:**
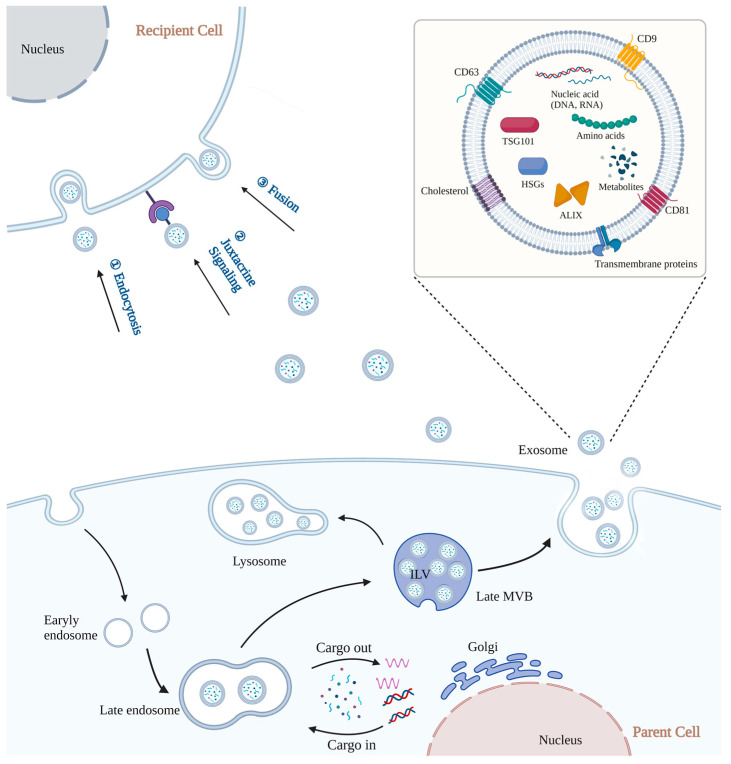
The biogenesis and cellular uptake modes of natural exosomes. During inward budding of the endosomal membrane, cargos are sorted into the forming vesicles through specific protein complex. By fusion of the MLVs with plasma membrane, exosomes are secreted into extracellular environment. Three ways of exosome cellular uptake: (1) endocytosis; (2) juxtacrine signaling; (3) Fusion.

**Figure 2 pharmaceutics-15-00598-f002:**
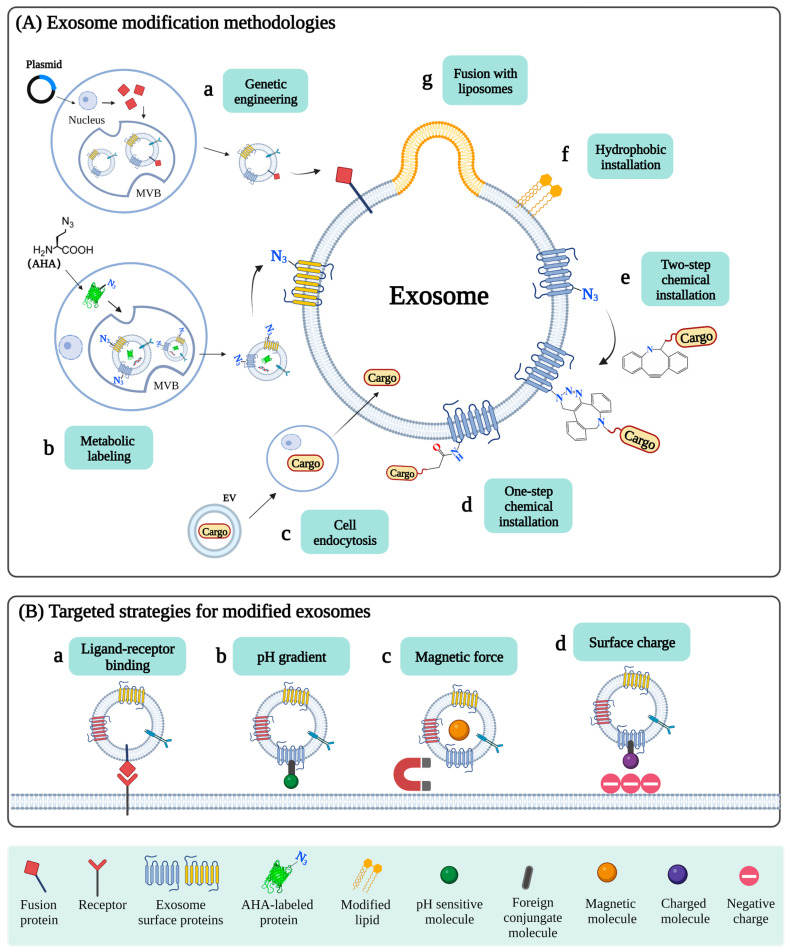
Overview of exosome modification and targeted delivery. (**A**) Exosome modification methodologies. There are a number of approaches to modifying exosomes to achieve targeted delivery in practical applications. Exosomes can be engineered before they are isolated, by expressing fusion proteins on the exosomal surface (**a**). In addition, exosomes bearing functional groups, such as azides, can be produced by incubating the donor cells with azide-labeled metabolites (**b**). The azide-labeled exosomes can be further functionalized through either physical or chemical modification. Exogenous material can be introduced to exosomes by the endocytosis and exocytosis of the parent cells (**c**). Chemical installation of simple molecules can be achieved by direct reaction with amino acid side chains of exosomal membrane proteins (**d**). To install complicated molecules, azide groups are first introduced to the exosomal membrane. The cargos can be conjugated to dibenzobicyclooctyne (DBCO) and then introduced to the exosomal surface by strain-promoted azide-alkyne click reaction (SPAAC) (**e**). Physical modification can be applied by hydrophobic insertion (**f**) or fusion with liposomes (**g**). (**B**) Targeted strategies for modified exosomes. Engineered exosomes are guided to targeted cells through ligand-receptor-specific binding, magnetic force, pH gradient, and surface charge attraction.

**Table 2 pharmaceutics-15-00598-t002:** Methods of RNA loading.

Loading Methods	Principle	Benefits	Drawbacks	Refs
Incubation	Diffusion into the lipid bilayer of exosomes or exosome-generating cells	Simple operation	Low loading efficiency, cytotoxicity	[88,89]
Transfection	Gene editing	Convenient operation, high efficiency, stability	Cost consuming, cytotoxicity, lack of specificity	[100,101,110]
Physical approach (Electroporation)	Create small holes on the membrane via electric field	Rapid operation, high efficiency	Membrane damage, cargo aggregation	[103,104]
Physical approach (Sonication)	Create small holes on the membrane via shear force	The rapid operation, high efficiency	Membrane damage, cargo aggregation	[105]
Physical approach (Dialysis)	Concentration/pH-depended permeation	Simple operation	Poor cellular uptake, cargo aggregation	[106]

**Table 3 pharmaceutics-15-00598-t003:** Application of RNA-loaded exosomes in cancer therapy.

Type of RNA	The Source of Exosomes	The Research Content	Ref.
mRNA	HEK-293 and dendritic cells	Exosomes that loaded HChrR6-encoding mRNA (convert prodrug (CNOB) into a drug (MCHB)) were delivered to HER2^+^ human breast cancer cells	[135]
	HEK-293 and dendritic cells	CB1954 was used as a prodrug to treat breast cancer similarly to enhance safety.	[136]
	Red blood cells (RBC)	Cas9 mRNA and guide RNA were loaded into exosomes and then target the human miR-125b-2 locus to treat leukemia.	[137]
	HEK293T, U87-MG, U251-MG and E98 cells	mRNA-containing exosomes transferred tumor suppressor genes into glioma cells and inhibit the growth of tumors.	[114]
	HEK-293T cells	Exosomes containing CD-UPRT mRNA were released to promote the conversion of the prodrug 5-fluorocytosine (5-FC) to 5-fluorouracil (5-FU) within tumor cells to treat Schwannoma.	[138]
miRNA	HT-29 and SW480 cells	miR-375-loaded tumor exosomes were generated and effectively targeted metastatic colorectal cancer tissues to control the development of tumors.	[141]
	Bone-marrow-derived mesenchymal stem cells	LNA (locked nucleic acid)-modified anti-miR-142-3p oligonucleotides were loaded to downregulate the level of miR-142-3p and miR-150 in 4T1 and TUBO breast cancer cell lines	[142]
	HEK-293T cells	Engineered exosomes encapsulated 5-FU and miR-21 inhibitor oligonucleotide (miR-21i) together to target 5-FU-resistant colorectal cancer cell line HCT-116^5FR^	[143]
siRNA	Fibroblast-like mesenchymal cells,	siRNA-loaded exosomes were isolated and modified to treat pancreatic cancer.	[24]
	mesenchymal stem cells (MSC), and HEK-293T cells	PLK-1 siRNA-loaded exosomes were generated to target bladder cancer tissues, resulting in cell apoptosis by downregulating the expression of PLK-1.	[145]
	Bone marrow mesenchymal stem cells (BM-MSCs)	Exosomes were isolated to deliver siRNA specific to Grp78 together with Sorafenib in sorafenib-resistant cancer cells	[146]
	HCT116, HEK-293T, and sw480	iRGD-modified exosomes delivered siCPT1A to silence the gene expression of an important enzyme promoting drug resistance namely fatty acid oxidation (FAO).	[147]
	A549 and H460 cells	PD1 modified exosomes loaded siRNA to target A549 and H460 cells.	[148]
	Bone marrow mesenchymal stem cells (BM-MSCs)	Oxaliplatin (OXA) and gal-9 siRNA were loaded together into exosomes to induce tumor-suppressive immune reactions.	[149]
circRNA	GC cell lines (MGC-803 and HGC-27)	circDIDO1-loaded, RGD-modified exosomes (RGD-Exo-circDIDO1) were generated to modulate the miR-1307-3p/SOSC2 axis.	[153]
chimeric RNA	Dendritic cells	Chimeric RNA encoding A-Pas, a cancer-specific chimeric protein was loaded into exosomes to construct an antitumor vaccine to treat esophagus cancer.	[154]
lncRNA	Osteosarcoma cell	c(RGDyK)-modified and MEG3-loaded exosomes were prepared to treat osteosarcoma.	[155]

## Data Availability

Not applicable.

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
