# Peer review of "Exosome-Based Carrier for RNA Delivery: Progress and Challenges"

_pharmaceutics, 2023, doi:10.3390/pharmaceutics15020598_

Round 1
Reviewer 1 Report
The review by Lu et al. is intended to inform readers on the progress made on exosomes as carriers for RNA therapeutics / RNA vaccines / RNA delivery into cells.
In my opinion, a central aspect of this review should be how to load exosomes with RNA. The authors state that “the most straightforward way to load RNA is incubation”. Incubation is also presented as a method in table 1, with the principle of “diffusion into the lipid bilayer of exosomes or exosome-generating cells”. But RNA cannot diffuse into exosomes spontaneously so this section is misleading. Reference 63, cited in this section, use a cholesterol-anchored miRNA to load exosomes / vesicles with RNA by incubation, but this is not discussed. In fact, the authors of reference 63 tried incubating exosomes with non-modified miRNAs and loading was very inefficient. Table 1 also cites reference 70, which is a review on vesicle characterization techniques that does not cover RNA loading.
The next paragraph mentions transfection and that “the desired RNA can be sorted into exosomes in donor cells through natural mechanisms”. But these natural mechanisms are not discussed. There is plenty of literature supporting different models of how cells load RNAs into exosomes, but these aspects are mostly not covered in the review.
The review also lacks a deep discussion on the advantages of using exosomes rather than liposomes or lipid nanoparticles for RNA delivery. There are some mentions of prolonged times in the circulation, but several key papers are either not cited or not discussed. For instance, reference 102 shows the role of CD47 in avoiding clearance by macrophages and reports profound difference in RNA delivery between iExosomes and iLiposomes, but these relevant aspects of the work are not discussed.
Exosomes are said to be taken up by recipient cells randomly (page 7), but this statement disregards potential effect of exosomal integrins in dictating tissue retention and enrichment (Hoshino et al. 2015). Pharmacokinetics studies of vesicles in primates also suggest preferential association with specific cell types (Driedonks et al. 2022). Throughout the manuscript, the term “exosome” is used for different classes of extracellular vesicles. For instance, the direct pathway (2) in Figure 1 actually corresponds to the term “microvesicles” or “ectosomes”, rather than exosomes as per MISEV guidelines.
Exosomes are said to contain 4563 proteins, 1639 mRNAs, 194 lipids and 764 miRNAs. This is misleading as the composition of extracellular vesicles is highly context-dependent. When discussing proteins commonly found in exosomes, CD81, CD82, CD37 and CD63 are introduced, ignoring CD9, which is present in Figure 1 as an exosome-enriched membrane protein. Figure 3 does not add much information to the manuscript, rather than presenting several human organs and diseases.
There are several typos in the manuscript, like “RNA-loaded RNA” in page 6, or weird phrases like “provoking cells with electrical motivations” just afterwards. More concerning, several references are wrongly listed. For instance, Alvarez-Erviti et al. 2011 appears as reference 86 and as reference 100, and is cited as reference 61 on page 16, with reference 61 being Valadi et al. 2007. Having a curated reference list is key in any paper, but especially in a review. Additionally, reviews are often cited rather than primary literature in support of relevant and specific statements.
Author Response
Thanks for your suggestions. Our response has been attached. Please see the attachment.

Reviewer 2 Report
The review article “Exosome-based carrier for RNA delivery: progress and challenges” is an interesting review article, provided with updates in the field. However, the manuscript is hampered by several wrong references and improper citations throughout the manuscript. For example, except one reference in the paragraph 2 of introduction, all the references are wrongly or improperly cited. Same trend is seen throughout the manuscript. This need to be strictly evaluated.
Following are my suggestions to improve the manuscript.
1. In the abstract, the following sentence “Exosomes have been exploited as a promising vehicle for drug delivery due to its nanoscalesize excellent stability, high biocompatibility, low immunogenicity, and ideal targeting capabilities” is an hyped statement based on the available original research. As per the statement exosomes are the best for drug delivery. However, we still need a lot of proof to prove the statement. So, this sentence should be modified.
2. All the sentences in the second paragraph of the introduction either have a review article reference or wrong references. The entire paragraph should be rewritten with essential original article references while making critical statements.
3. In the body of the review, the generation, composition, function and isolation of exosomes can either removed or drastically minimized as we have several review articles focusing on these components. It is better to focus only on RNA part for the benefit of readers. With the inclusion of these parts, the focus of the review will be lost.
4. Authors need to provide a complete paragraph (in detail) on future directions of the exosome-based carrier for RNA delivery. This need to be more specific to RNA delivery rather than on large scale isolation and purification related concerns.
Author Response

(The authors gave the same response as above.)

Reviewer 3 Report
Please check the attached file for comments and edits ! Just click on the comment mark.

Author Response

(The authors gave the same response as above.)

Round 2
Reviewer 1 Report
The manuscript has greatly improved. It now reads better and my most serious concerns were addressed. I still think the review lacks references to recent studies showing the pharmacokinetics and biodistribution of injected vesicles (relevant to the topic), but this is a minor point.
I read the new version quickly but still noticed several errors. For example, the new legend of Figure 1 describes fusion of ILV with the plasma membrane, but ILVs do not fuse with the PM (MVB do). Then, the legend describes "three ways of exosome uptake" but repeats "fusion" in (1) and (3).
Other typos: "Target Delivery o4 Exosome surface modification" and the phrase "Techniques to artificially increase the contents of specific RNAs [104]" seems incomplete.
Careful proofreading of the whole manuscript, with special emphasize on matching references in the text and in the reference list is needed. I am not going to do that job for a third time, so I gently ask the authors and editors to pay special attention to this, if the review is going to be published.
Author Response
Thanks for you valuable suggestions.
Point 1: I still think the review lacks references to recent studies showing the pharmacokinetics and biodistribution of injected vesicles (relevant to the topic), but this is a minor point.
Response 1: I have briefly disscussed the pharmacokinetics of exosomes after injection and added a few cites in page2 line 71 as below.
"Though after intravenous injection, unmodified exosomes are likely to be cleared from circulation rapidly and mainly distributed to the liver, spleen, lung, and gastrointestinal tract[27,28], by genetic engineering methods like transfection, homing peptides or ligands on the exosomal membrane can be expressed, which improves targeting and therapeutic efficiency[29,30]"
Point 2: I read the new version quickly but still noticed several errors. For example, the new legend of Figure 1 describes fusion of ILV with the plasma membrane, but ILVs do not fuse with the PM (MVB do). Then, the legend describes "three ways of exosome uptake" but repeats "fusion" in (1) and (3).
Other typos: "Target Delivery o4 Exosome surface modification" and the phrase "Techniques to artificially increase the contents of specific RNAs [104]" seems incomplete.
Response 2: I had checked our manuscript and references thoroughly and had corrected several mistakes.
Reviewer 2 Report
Authors have done significant modifications
Author Response
Thanks for your suggestion. I have checked our manuscript and references thoroughly and had corrected several mistakes.